# Ruthenium-cobalt nanoalloys encapsulated in nitrogen-doped graphene as active electrocatalysts for producing hydrogen in alkaline media

Jianwei Su[1],[*], Yang Yang[1],[*], Guoliang Xia[1], Jitang Chen[1], Peng Jiang[1] & Qianwang Chen[1],[2]

The scalable production of hydrogen could conveniently be realized by alkaline water electrolysis. Currently, the major challenge confronting hydrogen evolution reaction (HER) is lacking inexpensive alternatives to platinum-based electrocatalysts. Here we report a high-efficient and stable electrocatalyst composed of ruthenium and cobalt bimetallic nanoalloy encapsulated in nitrogen-doped graphene layers. The catalysts display remarkable performance with low overpotentials of only 28 and 218 mV at 10 and 100 mA cm$^{-2}$, respectively, and excellent stability of 10,000 cycles. Ruthenium is the cheapest platinum-group metal and its amount in the catalyst is only 3.58 wt.%, showing the catalyst high activity at a very competitive price. Density functional theory calculations reveal that the introduction of ruthenium atoms into cobalt core can improve the efficiency of electron transfer from alloy core to graphene shell, beneficial for enhancing carbon–hydrogen bond, thereby lowing $\Delta G_{H^*}$ of HER.

[1] Hefei National Laboratory for Physical Science at Microscale, Department of Materials Science & Engineering & Collaborative Innovation Center of Suzhou Nano Science and Technology, University of Science and Technology of China, Hefei 230026, China. [2] High Magnetic Field Laboratory, Hefei Institutes of Physical Science, Chinese Academy of Sciences, Hefei 230031, China. * These authors contributed equally to this work. Correspondence and requests for materials should be addressed to Q.C. (email: cqw@ustc.edu.cn).

ydrogen energy, a renewable energy resource, has been intensely investigated as an ideal alternative to the conservative fossil fuels for its high gravimetric energy density, zero-emission and earth-abundance[1–3]. Despite its promising application prospect, the current industrial route of hydrogen production not only consumes the fossil fuels but also brings on the greenhouse gas $CO_2$ emission[4,5]. Thus, exploring a cleaner, more sustainable and efficient strategy is urgently needed. Currently, electrochemical water splitting, a non-fossil fuel-based technology, is evoking increasing interests and stimulating intense investigations to produce low-costing and high-pure hydrogen[6,7]. As the heart of the scalable hydrogen production, it is of great importance to develop highly efficient electrocatalysts to drive the hydrogen evolution reaction (HER)[8].

It is well known, electrochemical water splitting can be carried out in acidic or alkaline media[2,4]. Unfortunately, the acid electrolyzers are technologically and commercially hindered by the lacking of efficient and low-cost counter electrode catalysts in acidic electrolytes[2,9,10]. Consequently, tremendous efforts have been devoted to developing HER catalysts with high activity and stability in basic media on the basis of the available alkaline oxygen evolution electrocatalysts, aiming at accelerating commercialization of the basic electrolyzers for $H_2$-production.

Platinum, as 'the Holy Grail' of HER electrocatalysts, remains the best HER catalysts with nearly zero overpotential and excellent long-term durability[3,11–13]. Unfortunately, the widespread commercialization of Pt-based electrocatalysts are hindered by their scarcity and expensive price[14,15]. Therefore, it is of great significance to explore inexpensive alternatives for Pt electrocatalysts. To date, plenty of robust and efficient alternative catalysts have been reported aiming at replacing Pt-based electrocatalysts in alkaline media. Among these materials, transition metal (TM) -based catalysts, including Mo-based catalysts[2,4,9,16,17], Ni-based catalysts[6,10,18,19] and Co-based catalysts[14,20–22], have been proven to be competitive eletrocatalysts as a result of their high efficiency and low cost. In particular, the biphasic nanocrystalline Ni–Mo–N catalyst recently reported by Li's group exhibited amazing HER activity with low overpotentials of 43 mV in 1 M KOH and 53 mV in 0.5 M $H_2SO_4$, which is close to commercial Pt/C catalyst at the same mass loading of 1 mg cm$^{-2}$ (ref. 2). Although, some of the TM-based catalysts show high HER performance, but still inferior to the Pt-based catalysts in overpotential and durability, which would increase energy consumption and hence decreasing economic competitiveness[23,24]. Currently, the pure TM-based catalysts are unable to meet the requirement of replacing Pt-based electrocatalysts.

For decades, other cheaper platinum-group metals have been investigated for HER in view of their high similarity to Pt in chemical inertness[13,25–27]. In particular, ruthenium (42 \$ per oz) is more economically advantageous in price than the rest of Pt-group metals, such as Pt (992 \$ per oz), Pd (551 \$ per oz), Ir (500 \$ per oz) and so on[28]. As a matter of fact, Ru has evoked special attention as a top oxygen evolution electrocatalytic material[29]. However, it is not very active for HER in basic media in previous works[30–33]. It is shown that alloying noble metals with other TMs, with the amount of noble metal even decreased by up to an order of magnitude, is a major route to prepare highly efficient catalysts with balance of good cost-competitiveness. Moreover, previous studies have proven that the chemical properties of bimetallic surfaces could be modified by the combined changes in the average energy of the surface d-band and in the width of the d-band due to the cumulative strain and ligand effects originated from the formation of heteroatom bonds and the alteration of the bond length[8,34]. Therefore, in addition to lowering the material cost, the noble-transition bimetallic alloys

could remarkably boost the electrocatalytic activities contributed by the shift of charge distributions and the resulting modification of surface properties during the formation of alloys[26,35,36].

In this work, a Ru-based electrochemical catalyst for HER was developed by alloy Ru with TM Co. Pervious work of our group and Bao's group have shown that a metal core coated with a carbon shell, especially N-doped graphene, can simultaneously promote the HER activity through the synergism and enhance the stability due to the protection from carbon cage[8,37]. Inspired by this, a metal-organic frameworks (MOFs) -assisted strategy was adopted for the preparation of RuCo nanoalloys encapsulated in nitrogen-doped graphene layers (RuCo@NC). We adopted an in situ method that consists of the one-step annealing of Ru-doped Prussian blue analogues, which are ideal nitrogen-rich precursors composed of metals as nodes and CN$^-$ groups as linkers, for the fabrication of alloy materials wrapped in N-doped carbon[8,26,38–41]. The RuCo@NC hybrid material shows an unprecedented high electrocatalytic performance towards HER in basic conditions, even superior to the commercial Pt/C catalysts.

## Results

**Synthesis and characterization of RuCo@NC catalyst.** Obviously, Fig. 1 illustrates the synthetic route and model of the RuCo@NC hybrids. The $Co_3[Co(CN)_6]_2$ MOF precursor particles, designated as S-0-MOF, were synthesized according to our previous studies[41,42]. As revealed by the field-emission scanning electron microscopy (FESEM) and transmission electron microscopy (TEM) images (Supplementary Fig. 1a,b), the morphologies of the S-0-MOF particles were truncated nanocubes which have very narrow diameter distributions with a mean diameter of ∼100 nm. In Fig. 1b, the Ru-doped $Co_3[Co(CN)_6]_2$ precursor were obtained via an ion-exchange reaction in the liquid phase. In brief, various $RuCl_3$ solution was added into the $Co_3[Co(CN)_6]_2$ solution, and the corresponding products were hereinafter designated as S-1-MOF, S-2-MOF, S-3-MOF, S-4-MOF, S-5-MOF and S-6-MOF, respectively. After adding Ru source, $Ru^{3+}$ ion diffused into the open framework of $Co_3[Co(CN)_6]_2$, which triggered an ion-exchange reaction between $Ru^{3+}$ and $Co^{3+}$ while maintaining intrinsic framework structure[43,44]. The FESEM and TEM images of the obtained Ru-doped precursor particles were illustrated respectively in Fig. 2a,b and Supplementary Fig. 2–6a,b, exhibiting that the as-prepared noble metal-doped $Co_3[Co(CN)_6]_2$ inherited the nanocubic morphology with an average diameter of ∼100 nm. The corresponding X-ray diffraction patterns are shown in Fig. 2c. The patterns of the samples all showed no additional reflections except a series of Bragg reflections corresponding to the diffractions from the $Co_3[Co(CN)_6]_2$ (JCPDS No. 77 − 1161). The unchanged X-ray diffraction patterns of Ru-doped MOFs suggest that $Co_3[Co(CN)_6]_2$ could maintain the intrinsic framework structure during the Ru doping process, which is in good agreement with the unchanged morphologies of the precursors discussed above. The X-ray photoelectron spectroscopy (XPS) spectra of S-4-MOF were revealed in Fig. 2d and Supplementary Fig. 9. Obviously, the Ru 3p peaks at 463.3 and 485.8 eV were, respectively, assigned to Ru 3p 3/2 and Ru 3p 1/2 for Ru(III) species, conforming that $Ru^{3+}$ have taken the place of some $Co^{3+}$ ions. Essentially, the Ru doping process was carried out in the MOF framework structure via an ion-exchange reaction[43].

In this work, the electrocatalysts were synthesized via an 'in situ' formation process without adding any other carbon sources. $Co_3[Co(CN)_6]_2$ is composed of CN$^-$ groups as linkers and TM Co as metallic nodes, which is an ideal precursor for the

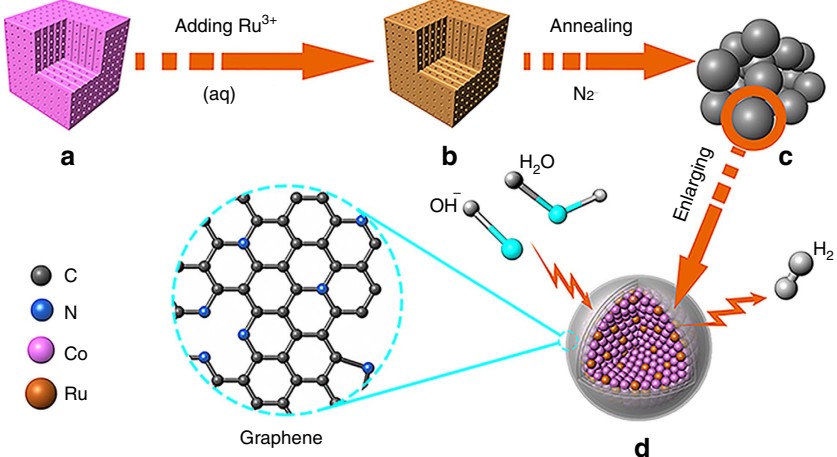

**Figure 1 | Schematic illustration of the synthetic route and model of the RuCo nanoalloys encapsulated in nitrogen-doped graphene layers.** (**a**) a nanocube of $Co_3[Co(CN)_6]_2$ precursor, (**b**) a nanocube of Ru-doped $Co_3[Co(CN)_6]_2$, (**c**) an aggregate of RuCo alloys encapsulated in graphene layers and (**d**) enlarged model of RuCo nanoalloy encapsulated in nitrogen-doped graphene layers as an electrocatalyst towards hydrogen evolution reaction in alkaline media.

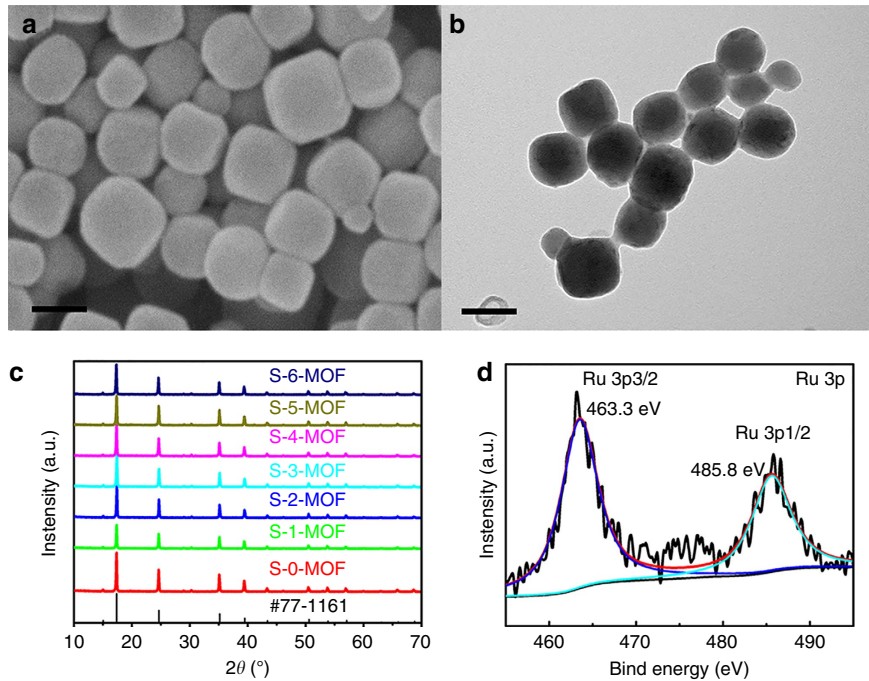

**Figure 2 | Ru-doped MOF $Co_3[Co(CN)_6]_2$ MOF precursor particles.** (**a,b**) FESEM and TEM images of the as-prepared S-4-MOF. Scale bars, 100 nm. (**c**) The X-ray diffraction patterns of S-0-MOF, S-1-MOF, S-2-MOF, S-3-MOF, S-4-MOF, S-5-MOF and S-6-MOF. (**d**) The XPS result of the Ru 3p spectrum enlarged from Supplementary Fig. 9a.

preparation of composite materials of TM-based material with highly N-doped carbon[39]. Specifically, the corresponding RuCo@NC hybrids were synthesized via one-step annealing of the obtained Ru-doped MOFs, hereinafter marked as S-0, S-1, S-2, S-3, S-4, S-5 and S-6, respectively. As illustrated in Fig. 1c,d, the nanocubic MOF precursors were directly carbonized at 600 °C under a nitrogen flow without adding any other carbon sources. The resulting product is composed of the bimetallic RuCo nanoalloys encapsulated in nitrogen-doped graphene layers. During the annealing process, Ru and Co atoms from the precursor would form the bimetallic RuCo nanocrystals, meanwhile some remaining $CN^-$ group linkers would transfer into nitrogen-doped graphene layers. As can be seen in Fig. 1d, the formation of RuCo nanoalloys will be coated speedily by the *in situ* formed N-doped graphene layers, which is able to efficiently avoid the agglomeration of the inside alloy particles to provide more eletrocatalytic active sites which are beneficial to HER activity and long-term corrosion protection to enhance the catalytic durability.

All the obtained samples embraced an irregular particle-like morphology according to the FESEM results (Fig. 3a and Supplementary Fig. 2–6d). As revealed in the TEM images (Fig. 3b, Supplementary Figs 2–6c,e,f and 7 and Supplementary Table 1), the irregular products were composed of small alloy

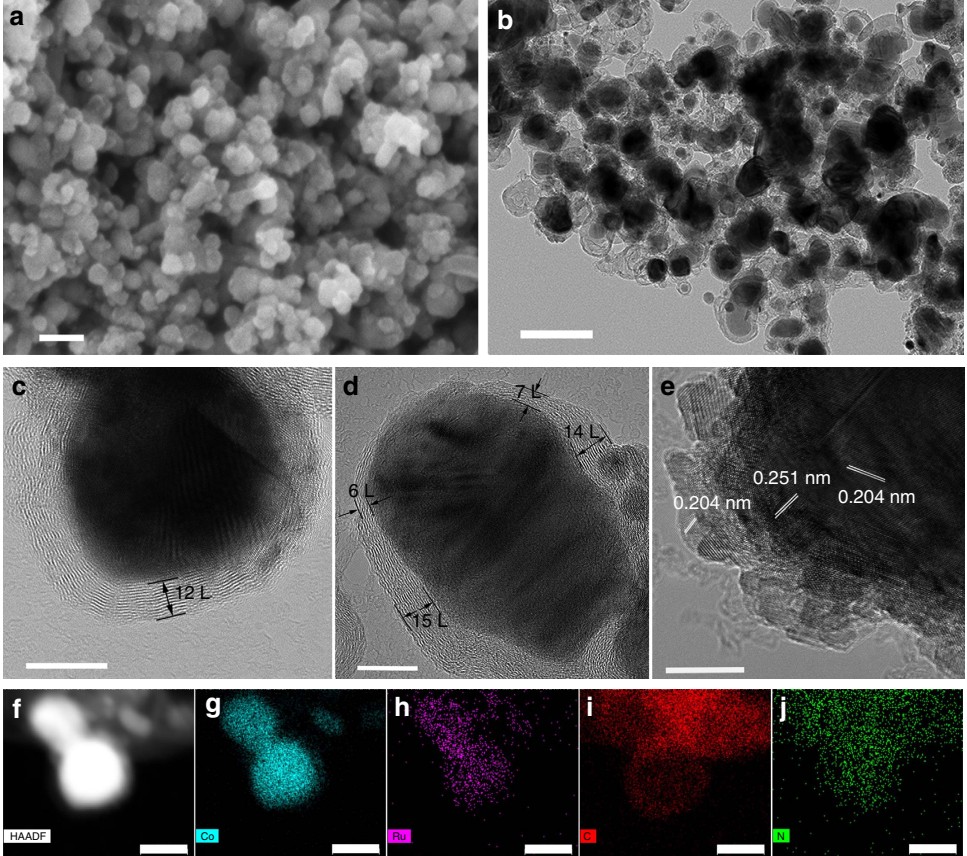

**Figure 3 | SEM and TEM characterization of RuCo@NC hybrid.** (**a,b**) The FESEM and TEM images of S-4. Scale bars, 100 nm. (**c–e**) HRTEM images of S-4. Scale bars, 10 nm. (**f–j**) HAADF-STEM image and corresponding EDX maps of S-4 for Co (**g**), Ru (**h**), C (**i**) and N (**j**), respectively. Scale bars, 20 nm.

particles with a mean diameter of ∼30 nm encapsulated carbon layers. The high-resolution TEM (HRTEM) images (Fig. 3c,d) showed that the small alloy particles were coated with N-doped graphene layers. Most of the graphene layers were ∼6–15 layers thick. However, as shown in Supplementary Fig. 8, a large proportion of graphene shells consisting of 1–5 layers still existed, which are good active sites for HER. The thin graphene shells were beneficial in electron transfer from the alloy core to the shell and hence improved the catalytic activity on the graphene surface[8,37]. Besides, Fig. 3e clearly revealed that lattice fringe spaces of 0.251 and 0.204 nm are, respectively, consistent with the (111) and (220) planes of the cubic Co phase or the (002) and (220) planes of the hexagonal Co phase. While, no lattice fringe spaces of metallic Ru were found, implying the formation of RuCo alloy which kept the crystal structure of metallic Co. As shown in Fig. 3f–j, the images of elemental mapping from energy filtered TEM showed that the Co and Ru elements (5.69 wt.% Ru and 94.31 wt.% Co) were uniformly distributed in the inner particles and surrounded by the C and N elements, further confirming the formation of RuCo nanoalloys encapsulated in nitrogen-doped graphene layers.

The corresponding X-ray diffraction patterns of RuCo@NC were shown in Fig. 4a. Similar diffraction features of a broad and weak peak of C (002) were detected, confirming the existence of carbon layers. Besides, the patterns of all the samples showed no additional reflections except a series of Bragg reflections corresponding to the diffractions from the hexagonal structure Co (JCPDS card no. 05 − 0727) phase and face centred cubic (FCC) Co (JCPDS card no. 15-0806) (refs 7,26), suggesting that Ru atoms were *in situ* dissolved in metallic Co to form RuCo

alloy, which is in good agreement with the value determined by the HRTEM observations discussed above.

The Raman spectra of S-4 were shown in Fig. 4b. The product displayed three Raman peaks locating at ∼1,349, 1,583 and 2,703 cm$^{-1}$, which corresponded to the D, G and 2D bands, respectively. The high $I_D/I_G$ band intensity ratio of S-4 indicated the generation of large amounts of defects, suggesting that a large amount of N atoms were doped in the graphitic carbon layers. Moreover, the second-order band is broad and weak, implying that the coated carbon is thin graphene with several layers[45].

The XPS spectra of S-4 were illustrated in Supplementary Fig. 10a. The nitrogen content of S-4 was ∼3.51 atom%. The high-resolution N1s spectrum of S-650 (as shown in Fig. 4c) can be deconvoluted into four individual peaks assigned to pyridinic-N (398.5 eV), pyrrolic-N (399.6, 400.9 eV) and quaternary-N (401.6 eV), respectively[8,46]. Importantly, two pyrrolic-N-binding energies are observed here. This phenomenon might result from the energy shift which was induced by the interaction between some pyrrolic-N and metal atoms[47]. Therefore, the percentages of pyridinic-N, metal-N, pyrrolic-N and quaternary-N were 40%, 17%, 27% and 16%, respectively. Besides, XPS investigation of the Co 2p spectrum revealed the presence of two distinct chemical species: Co$^0$ and Co$^{II}$ species peaks, indicating the existence of Co$^{II}$ originated from surface oxidation of metallic Co. The existence of metallic Ru was also confirmed by the weak peak at 462.2 eV (Supplementary Fig. 10c), suggesting that the Ru content was very tiny in the RuCo alloys with good balance of its cost.

The specific surface area and pore size distribution were obtained by N$_2$ adsorption/desorption isotherms. As can be seen in Fig. 4d, a type-II isotherm with a H3-type hysteresis loop is

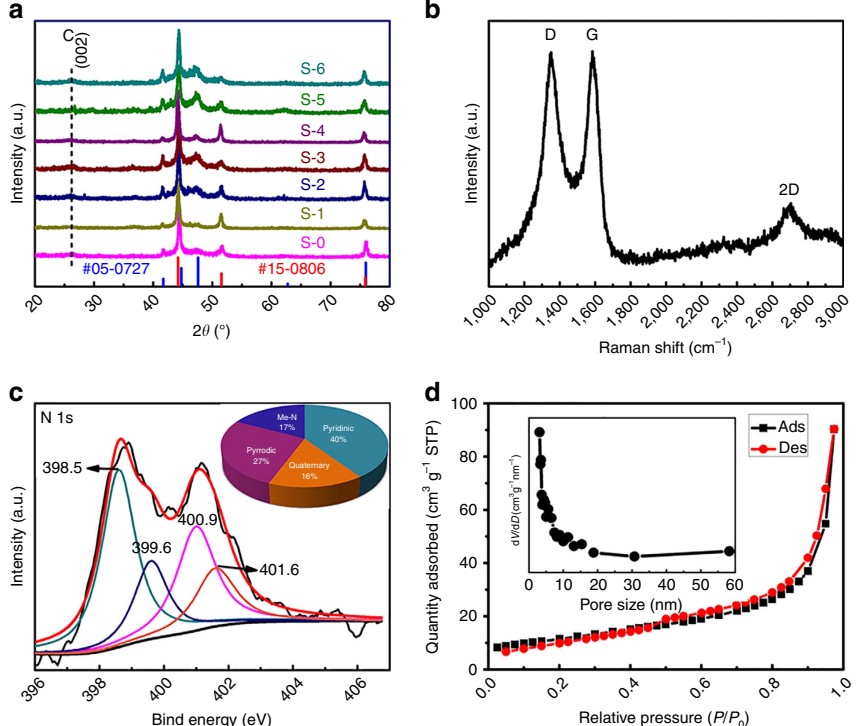

**Figure 4 | Structural analysis of RuCo@NC hybrid. (a)** The X-ray diffraction patterns of S-0, S-1, S-2, S-3, S-4, S-5 and S-6, respectively. **(b)** Raman spectrum of S-4, **(c)** The XPS result of the N1s spectrum enlarged from Supplementary Fig. 10a, **(d)** $N_2$ adsorption–desorption isotherm and pore size distribution plot (inset) of S-4.

obtained, which is characteristic of mesoporous non-rigid aggregates[48]. Obviously, S-4 have a specific surface area of $41.365 \, m^2 \, g^{-1}$. The pore size distribution (insert image) revealed a strong and narrow distribution centred at $\sim 3 \, nm$. The other samples showed the similar adsorption/desorption isotherm curves and pore size distribution in Supplementary Fig. 11. The BET surface area were listed in Supplementary Table 1. These results were in good agreement with the value determined by the morphology observations discussed above.

**Electrochemical characterization for HER catalysis.** The electrochemical catalytic activities of the annealed samples for HER were recorded by a typical three-electrode electrochemical cell in $N_2$ saturated 1 M KOH electrolyte. The HER performance was evaluated by observing the overpotential versus the reversible hydrogen electrode (RHE) at $10 \, mA \, cm^{-2}$, which is the current density expected for a 12.3% efficient solar water-splitting device[15]. As revealed in Fig. 5a, the undoped S-0 catalyst exhibited an excellent HER activity but far inferior to the doped counterparts of RuCo alloy encapsulated in N-doped graphene layers, proving that alloying TMs with less noble metals played a key role in boosting the electrocatalysis activity with good balance of the cost. The polarization curves also showed that the S-4 catalyst had the highest HER activity among seven catalysts with an overpotential of only 28 mV without IR-correction. As illustrated in Table 1, the trend in the overpotentials was found to be S-4 (28 mV) < S-3 (67 mV) < S-2 (83 mV) < S-1 (91 mV) < S-0 (300 mV), suggesting that the activity was increased with the increasing amount of Ru. Interestingly, the activity would decrease with the further increase of Ru amount. Besides, the similar trend of overpotentials could also be observed reaching a higher current density of $100 \, mA \, cm^{-2}$ (Table 1 and Supplementary Fig. 12b). To gain further insight into the

activity of S-4, the inner metal of S-4 was etched via 1 M HCl solution. Obviously, S-4 retains the same morphology, but a part of its metallic cores are removed after the acid leaching (Supplementary Fig. 13a). Besides, the HCl solution became pink in Supplementary Fig. 13c, suggesting that just a part of metallic Co was corroded and dissolved into the solution. It was also testified by the neglectable Ru concentration (0.46 wt.% Ru in Co and Ru) of ICP. It was shown in Fig. 5d and Supplementary Fig. 13d that both of the overpotentials reaching current densities of 10 and $100 \, mA \, cm^{-2}$ increased after etching, revealing that the metallic Co in S-4 was indispensable and had a key role in HER activity. It was also found that S-4 with lower Ru content (3.58 wt.% Ru in RuCo alloy, obtained from ICP) showed better activity than the etched counterpart with higher Ru content (17.7 wt.% Ru in RuCo alloy, obtained from ICP), which was in good agreement with the above- mentioned trend of overpotentials. All these findings revealed that S-4 (Ru, 3.58 wt. %, obtained from ICP) hold the best alloy structure for HER among RuCo alloys.

The electrochemically active surface area of samples was estimated using a simple cyclic voltammetry (CV) method[9,49]. Due to the unknown capacitive behaviour (Cs) of the RuCo alloy electrode especially with N-doped graphene shell, we can safely estimate relative surface areas of seven samples, since the double-layer capacitance (Cdl) is expected to be linearly proportional to effective active surface area for samples with similar composition and this method was also employed in previous study[8]. The results in Supplementary Fig. 14 suggested that with the increasing density of catalytically active sites of samples also follow the similar trend of the catalytic performance. A series of activity normalization of special activity (SA) and active site activity (ASA), taking into account of the surface area and active site concentration respectively at overpotential of 100 mV, were shown in Supplementary Fig. 16 (ref. 50). Especially, the

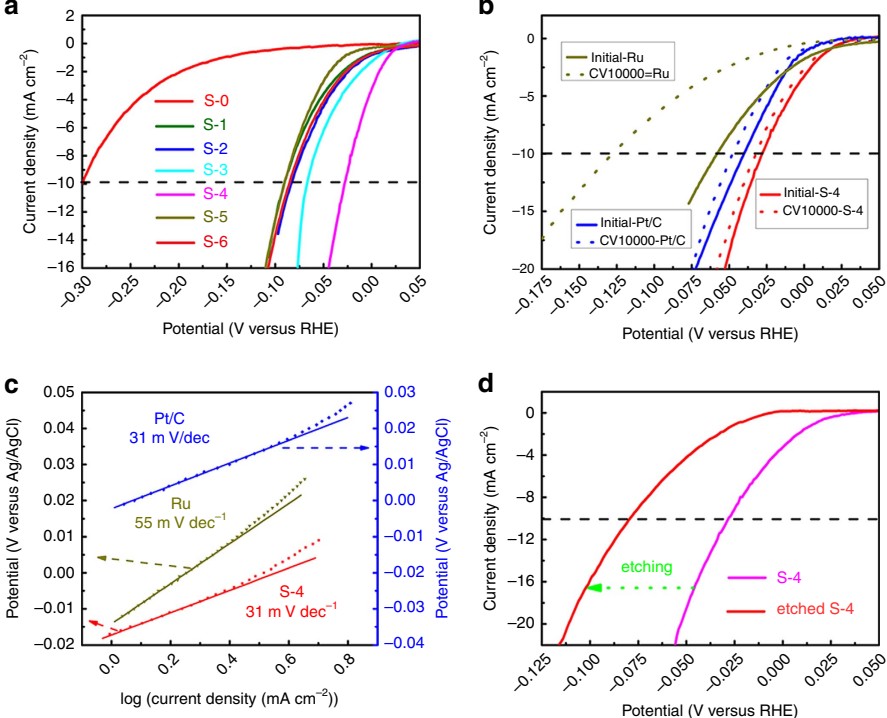

**Figure 5 | Electrocatalytic HER performance test of catalysts in $N_2$ saturated 1 M KOH solution.** (**a**) HER polarization curves of RuCo@NC samples with the same mass loading, (**b**) HER polarization curves of S-4, Ru and Pt/C with the same mass loading and durability test after 10,000th cycles, (**c**) the Tafel plots of S-4, Ru and Pt/C, (**d**) HER polarization curve of etched S-4 by 1 M HCl.

| Table 1 | Comparison of HER activity of different catalysts. | | | | | | | | |
|---|---|---|---|---|---|---|---|---|---|
| **Sample** | **S-0** | **S-1** | **S-2** | **S-3** | **S-4** | **S-5** | **S-6** | **Pt/C** | **Ru** |
| $\eta$@10 (mV) | 300 | 91 | 83 | 67 | 28 | 91 | 85 | 40 | 58 |
| $\eta$@100 (mV) | | 338 | 322 | 291 | 218 | 281 | 284 | 287 | 462 |
| Ru (wt.%) | 0 | 1.858 | 2.316 | 3.074 | 3.58 | 4.00 | 4.234 | 20 (Pt) | 99.9 |

HER, hydrogen evolution reaction.

S-4 catalyst exhibited a SA of 0.707 mA cm$^{-2}$, which are much higher than other catalysts. Besides, the result also suggests that the SA of samples increased with the increasing catalytic active in general, suggesting the effect of surface area is a negligible factor on activity. Due to the unknown capacitive behaviour ($C_s$) of the RuCo alloy electrode especially with N-doped graphene shell, we carried out activity normalization of ASA*$C_s$. Especially, the S-4 catalyst exhibited an ASA*$C_s$ of 1.242 mA cm$^{-2}$, which are much higher than other catalysts with the similar value of ASA*$C_s$. Besides, the electrochemical impedance spectroscopy images (Supplementary Fig. 15) showed that the Co@NC has a biggest semicircle radius, indicative of a slightly higher charge-transfer impedance than of S-1 (8.53 Ω), of S-2 (8.55 Ω), of S-3 (8.58 Ω), of S-4, of S-5 (8.53 Ω) and of S-6 (8.47 Ω) (refs 51,52). Therefore, alloying Co with small amount of Ru could lead to a higher charge-transfer rate and more facile catalytic kinetics toward HER.

The commercial Ru powders catalyst (Ru: 99.9 wt.%) and the commercial Pt/C catalyst (Pt: 20 wt.%) were also measured as references. Surprisingly, the obtained S-4 even exhibited better activity than Ru catalyst (58 mV) and Pt/C catalyst (40 mV) (Fig. 5b). Moreover, the fast HER kinetics also enabled the S-4 catalyst to reach a high current density of 100 mA cm$^{-2}$ at $\eta$ as low as 0.218 V, outperforming the other HER catalysts

including Pt/C (Supplementary Fig. 12a and Table 1). It is also the best one among some catalysts in the recent reports towards HER in basic media (Table 2)[1,2,4,6,9,17–19,21,22,25,53–56]. The Tafel slope (Fig. 5c) of the S-4 sample was 31 mV dec$^{-1}$, which was as same as the value of Pt/C, implying a rapid HER rate and a Tafel–Volmer mechanism with electrochemical desorption of $H_2$ as the rate-determining step in the HER process[57]. While, the Tafel slope was 55 mV for the Ru powder catalyst, which was much larger and indicative of a typical Volmer−Heyrovsky mechanism with the Volmer step as rate-limiting step for HER[6,53]. The exchange current density ($j_0$) values were obtained from Tafel plots[29]. As read from Supplementary Fig. 17, the catalysts showed an increase in $j_0$ in the following order: S-4 ($10^{-2.48}$) > Ru powder ($10^{-2.74}$) > Pt/C ($10^{-2.94}$). Durability was another critical parameter to assess the electrocatalytic performance. The durability of S-4 was also evaluated by measuring polarization curves after 10,000 CV sweeps between −1.1 and −0.9 V (versus Ag/AgCl) at 100 mV s$^{-1}$. As illustrated in Fig. 5b, the polarization curve of S-4 after 10,000 cycles retained an almost similar performance to the initial test, just with the overpotential increased by 4 mV, which is also slightly superior to the durability of Pt/C (8 mV) as well as much better than Ru powder. The graphene shell and the alloying of Ru and Co made great contributions to protect the catalysts from corrosion during

**Table 2 | HER electrocatalysts in alkalic media reported recently.**

| Catalyst | electrode | Loading amount (mg cm$^{-2}$) | Electrolyte | Overpotential at 10 mA cm$^{-2}$ (mV) | Tafel plots (mV dec$^{-1}$) | Reference |
|---|---|---|---|---|---|---|
| S-4 | GHE | 0.275 | 1 M KOH | 28 | 31 | This work |
| NiO/Ni-CNT | GHE | 0.28 | 1 M KOH | 80 | 82 | Nat. Commun., 2014[6] |
| Ni/Ni$_x$P$_y$ | Ni foam | N/A | 1 M KOH | 130 (with iR-correction) | 58.5 | Adv. Funct. Mater., 2016[18] |
| NF-Ni$_3$Se$_2$/Ni | Ni foam | 8.87 | 1 M KOH | 203 (with iR-correction) | 79 | Nano Energy, 2016[19] |
| Pt$_{13}$Cu$_{73}$Ni$_{14}$/CNF@CF | CNF@CF | N/A | 1 M KOH | 148 ($\eta_5$) | 54 | ACS Appl Mater Interfaces, 2016[21] |
| Pd-CN$_x$ | GCE | 0.28 | 0.5 M KOH | 180 ($\eta_5$) | 150 | ACS Catalysis, 2016[25] |
| Mo$_2$C@N-C | GCE | 0.28 | 1 M KOH | 60 | N/A | Angew Chem Int Ed Engl, 2015[4] |
| Ni–Mo–N | GCE | 1 | 1 M KOH | 43 ($\eta_{20}$) | 40 | Nano energy, 2016[2] |
| MoO$_2$/CC | Carbon paper | 2.9 | 1 M KOH | 100 (with iR-correction) | 41 | Adv. Mater., 2016[9] |
| MoO$_x$/Ni$_3$S$_2$ /NF | Ni foam | 12 | 1 M KOH | 110 ($\eta_{15}$) | 90 | Adv. Funct. Mater., 2016[17] |
| CoO$_x$@CN | Ni foam | 0.42 | 1 M KOH | 232 | N/A | J. Am. Chem. Soc., 2015[56] |
| CoP$_2$/RGO | GCE | 0.285 | 1 M KOH | 88(with iR-correction) | 50 | J. Mater. Chem. A, 2016[53] |
| CoP/rGO-400 | GCE | 0.28 | 1 M KOH | 150 | 38 | Chem. Sci., 2016[54] |
| c-CoSe2/CC | Carbon cloth | N/A | 1 M KOH | 190(with iR-correction) | 85 | Adv. Mater., 2016[22] |
| N-Co@G | GCE | 0.285 | 0.1 M NaOH | 337 | N/A | ACS Appl Mater Interfaces, 2015[55] |
| WC-CNTs | Si wafer | N/A | 0.1 M KOH | 137 | 106 | ACS Nano, 2015[1] |

HER, hydrogen evolution reaction.
Some of the information was not specified in the literature and was estimated according to the data graphs.

cycling[8,37]. Besides, Ru (42 $ per oz) is further more economically advantageous in price than Pt (992 $ per oz)[28]. The Ru content of S-4 was 3.58 wt. % obtained from ICP in Table 1, which was much lower than the Pt content of 20 wt. % in Pt/C catalysts. Specific to the field of noble metal's cost, the S-4 was only 0.76% of commercial Pt/C in price. Therefore, the S-4 sample performed higher activity, durability and economic competitiveness than the commercial Pt/C catalysts, demonstrating that it had a great potential to be a substitution of Pt/C catalyst for HER in alkaline media.

**HER enhancement mechanism.** To investigate the origin of excellent activity of RuCo alloy encapsulated in nitrogen-doped graphene layers for HER, density functional theory calculations were carried out using software of Vienna *Ab Initio* Simulation Package. A graphitic carbon cage C240 encapsulated 55 metal atoms was used as the basic model of graphene-encapsulated alloys, which worked well in previous studies[37,58]. The detailed calculation information could be read in the Supplementary Note 1. In general, for HER performed in both acid and alkaline electrolyte, $\Delta G_{H^*}$ is one of the key descriptors in theoretical prediction of the activity for HER. Previous studies have used $|\Delta G_{H^*}|$ as a catalytic descriptor for HER and proposed the optimal value should be close to 0 (refs 22,28). Therefore, the $\Delta G_{H^*}$ of different models were calculated, including pure graphene (C$_{240}$), graphene doped with nitrogen atoms (C$_{239}$N$_1$), graphene-encapsulated Co (C$_{239}$N$_1$Co$_{55}$) as well as graphene-encapsulated RuCo alloys. In order to find the influence of Ru content in RuCo alloys, three established different models were abbreviated to Ru$_1$Co (C$_{239}$N$_1$Ru$_1$Co$_{54}$), Ru$_2$Co (C$_{239}$N$_1$Ru$_2$Co$_{53}$) and Ru$_3$Co (C$_{239}$N$_1$Ru$_3$Co$_{52}$), in which the metal atom ratio is comparable to experimental materials measured by ICP. The optimized adsorption structures are shown in Fig. 6, which exhibited the C atom next to the doping N atom in graphene shell is the active adsorption site for H*, which is also consistent with previous work[8,37,53]. As illustrated in Fig. 7, the nitrogen doping and the combination of metal Co with graphene can significantly

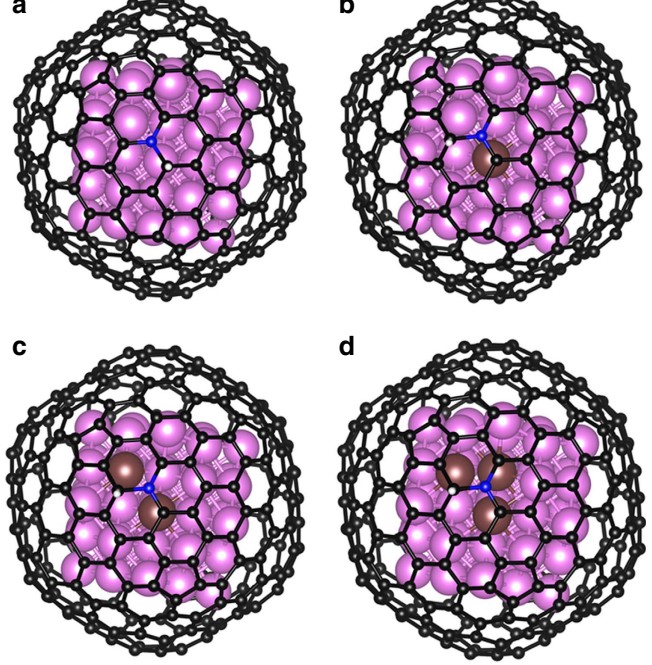

**Figure 6 | Optimized structures of H\* adsorbed on nitrogen-doped graphene-encapsulated Co and RuCo alloys.** (a) Pure Co model (C$_{239}$N$_1$Co$_{55}$), (b) Ru$_1$Co alloy model (C$_{239}$N$_1$Ru$_1$Co$_{54}$) and (c) Ru$_2$Co alloy model (C$_{239}$N$_1$Ru$_2$Co$_{53}$) (d) Ru$_3$Co alloy model (C$_{239}$N$_1$Ru$_3$Co$_{52}$). The black, blue, pink, brown and white balls refer to C, N, Co, Ru and H atoms, respectively.

reduce $\Delta G_{H^*}$, this result was consistent with our previous work[8]. Moreover, the introduction of Ru atoms into Co metal can further decrease $\Delta G_{H^*}$ and properly increasing Ru content will further reduce the value of $\Delta G_{H^*}$. As a result, Ru$_3$Co model

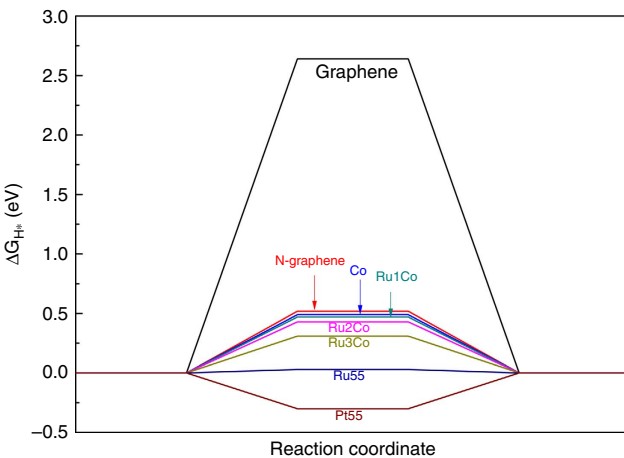

**Figure 7 | HER-free energy diagram.** $\Delta G_{H^*}$ calculated at the equilibrium potential of different models.

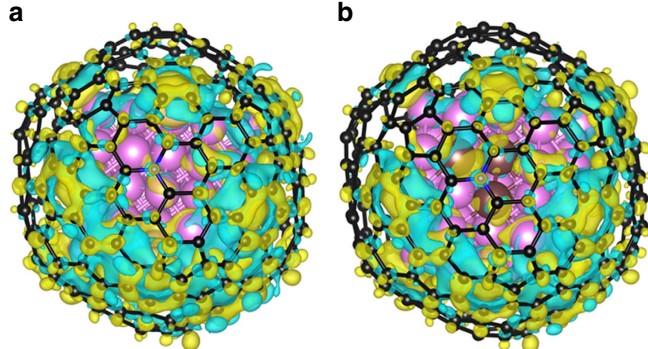

**Figure 8 | Calculated charge-density differences of different models.** (**a**) Co and (**b**) Co₃Ru models. The isosurface value of the colour region is $0.01\,e\,Å^{-3}$. The yellow and cyan regions refer to increased and decreased charge distributions, respectively.

exhibit the lowest $\Delta G_{H^*}$ (0.31 eV) among graphene-encapsulated models. We also calculated H* adsorbed on $Pt_{55}$ and $Ru_{55}$ metal cluster models. Besides, some possible H* adsorption sites such as top, bridge, hcp and hollow sites are taken into considerations in our cluster models (Supplementary Figs 19 and 20). It should be noted that, during optimization process the hollow site of $Pt_{55}$ and top site of $Ru_{55}$ are tended to be bridge and hollow sites, respectively, indicating the latter are more moderate adsorption sites. The calculated $\Delta G_{H^*}$ of different models and sites are illustrated in Supplementary Tables 2 and 4.

The calculated results show $Ru_{55}$ cluster exhibited even better $\Delta G_{H^*}$ than RuCo and $Pt_{55}$, which is not in good agreement with our previous experimental results based on purchased Ru powder. Unlike bulk metal, metal cluster and nano-sized metal particles have more high active sites such as facets, corners, edges. However, the Ru powder measured before was purchased from manufacturer and the size of them was in micron order as illustrated in Supplementary Fig. 18a. Besides, recent researches report nano metallic Ru-based electrocatalysts with very active performance in basic media[59,60]. Therefore, we synthesized nano-sized Ru nanoparticles (Supplementary Fig. 18b) by ourselves through $RuCl_3$ and $NaBH_4$ as Precursors. The performance of nano-sized Ru catalyst was even better than our S-4 at high overpotential (Supplementary Fig. 18d–f), which was in good agreement with our calculated cluster result. For Pt cluster, even its calculated result was as good as our CoRu models; however, the measured activity was inferior to our S-4 sample. Similar results also existed in a recently published research[59], Qiao and his co-workers have proved that, apart from the $\Delta G_{H^*}$, the water dissociation kinetics would also affect the overall reaction rate, especially under basic electrolyte. According to their calculated results, when the kinetics of water dissociation from the Volmer step is considered, Pt exhibits a significant higher energy barrier than Ru and others, indicating sluggish water dissociation during catalytic process. Therefore, from the kinetic viewpoint, conversely, Pt did not demonstrate as good catalytic performance as depicted by calculated $\Delta G_{H^*}$, which might explain the inferior measured activity than our CoRu alloys. We also tried to calculate energy barrier of water dissociation process for our CoRu alloys; however, due to the very large models ($C_{239}N_1Co_{52}Ru_3$ containing 295 atoms), it's very difficult and time-consuming to search for the exact transition state and energy barrier of ours. However, combing our calculated $\Delta G_{H^*}$ results with Qiao's result, it's sufficient to give a reasonable explanation. Therefore, both the experimental and the calculated results indicated that alloying Co with small amount

of Ru could obtain highly active eletrocatalyst comparable to pure Ru catalyst.

The charge-density difference of model Co and $Ru_3Co$ was also calculated and illustrated in Fig. 8. It is shown that electrons transferred from alloy core to C atoms of graphene shell, which was beneficial to enhance C–H bond thereby lowing $\Delta G_{H^*}$ of the model as a whole[37]. As shown in Supplementary Table 3, the exact number of transferred electron was calculated using bader charge analysis. Ru3Co model showed the largest number of transferred electrons ($5.91\,e^-$) among graphene-encapsulated models, therefore, $\Delta G_{H^*}$ of it was also the lowest among them. The calculated result is consistent with the experimental results, proving the excellent HER activity originates from Ru alloying with Co atoms as well as the unique structure derived from MOFs.

## Discussion

In summary, a novel RuCo alloy catalysts with high catalytic activity and stability for HER in basic solutions was prepared, which exceeds almost all the documented electrocatalysts including 20 wt.% Pt/C catalysts, hence enable it a cheaper alternative to Pt-based electrocatalysts for HER in basic media. The RuCo alloy structures with a low content of Ru provide it a huge economical advantage in price over commercial Pt/C catalysts. The density functional theory calculations indicate that RuCo alloy core can transfer more number of electrons to the graphene shell than pure Co metal core so as to enhance C–H bond, which will significantly decrease $\Delta G_H^*$, and thereby improve electrocatalysis activity. The results here show a new way for the development of high-performance HER electrocatalysts in alkaline media while reducing the cost of noble metal electrocatalysts.

## Methods

**RuCo@NC hybrids synthesis.** The $Co_3[Co(CN)_6]_2$ MOF was prepared according to our previous researchs[41,42]. The as-prepared $Co_3[Co(CN)_6]_2$ nanoparticles (25 mg) were dispersed in a 20 ml distilled water system under agitated stirring to get an absolutely homogeneous mixed solution, followed by the addition of 0.55, 1.1, 1.65, 2.2, 2.75 or 3.3 ml $RuCl_3$ solution ($0.01\,g\,ml^{-1}$). After agitated stirring for 10 h in dark, the brown products were collected and rinsed several times by distilled water, and finally dried under oven at 60 °C. The thermal decomposition of the MOF precursor was performed at 600 °C for 4 h under nitrogen atmosphere in the oven with a heating rate of $10\,°C\,min^{-1}$. The obtained sample should be kept in the vacuum drying oven.

**Nano Ru synthesis.** Briefly, 100 mg sodium borohydride ($NaBH_4$) was dissolved in a 20 ml distilled water system under agitated stirring to get an absolutely

transparent solution. And then the above solution was added into 20 ml RuCl$_3$ solution (5 mg ml$^{-1}$) slowly. After agitated stirring for 20 h, the black products were collected and rinsed several times by distilled water and ethanol, and finally dried under oven at 60 °C in the vacuum drying oven.

**Characterization.** The powder X-ray diffraction patterns of the samples were analysed with an X-ray diffractometer (Japan Rigaku D/MAX-γA) using Cu-Kα radiation ($\lambda = 1.54178$ Å) with $2\theta$ range of 20–80°. The morphology and size of all as-synthesized samples were characterized by a JEOL JSM-6700 M field-emission scanning electron microscopy and a scanning TEM (STEM; Talos F200X) with energy dispersive X-ray (EDX) spectroscopy. Raman spectrum was carried out using a LabRAM HR Evolution ranging from 1,000 to 3,000 cm$^{-1}$. XPS was measured on an ESCALAB 250 X-ray photoelectron spectrometer using Al Ka radiation. The specific surface and pore diameters were obtained from the results of N$_2$ physisorption at 77 K (Micromeritics ASAP 2020) by using the BET (Brunauer-Emmet-Teller) and BJH (Barrett-Joyner-Halenda) methods, respectively. The content of Co and Ru were obtained via the inductively coupled plasma-atomic emission spectrometer (ICP-AES) (Optima 7300 DV).

**HER electrochemical measurements.** The HER electrochemical perform measurements were performed in a three-electrode system on an electrochemical workstation (CHI 660D) in 1 M KOH electrolyte. Typically, 4 mg of catalyst and 30 μl Nafion solution (Sigma Aldrich, 5 wt.%) were dispersed in 1 ml of 3:1 v/v water/isopropanol mixed solvent by at least 30 min sonication to form a homogeneous ink. Then 5 μl of the dispersion (containing ~19.4 μg of catalyst) was loaded onto a glassy carbon electrode with 3 mm diameter (loading ~0.275 mg cm$^{-2}$). An Ag/AgCl (filled with 3 M KCl solution) electrode and a platinum wire were served as the reference electrode and counter electrode, respectively. All of the potentials were calibrated to a RHE. The working electrode was polished with Al$_2$O$_3$ powders with size ranging from 1 to 0.05 μm. Commercial Ru powders catalysts (99.9%, Aladdin) and commercial Pt/C catalysts (20%, Alfa Aesar) were used as a reference to evaluate the electrocatalytic performance of various samples. Linear sweep voltammetry with a scan rate of 2 mV s$^{-1}$ was conducted between $-1.1$ V and $-0.9$ V versus AgCl/Ag electrode into N$_2$ saturated 1 M KOH electrolyte. CV was conducted in 1 M KOH solution in the potential region from $-1.1$ V to $-0.9$ V versus AgCl/Ag electrode at a sweep rate of 100 mV s$^{-1}$ for 10,000 times to investigate the cycling stability.

**Data availability.** The authors declare that the data supporting data supporting the findings of this study are available within the article and its Supplementary Information files.

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

## Acknowledgements

This study was supported by the National Natural Science Foundation (NSFC, 21571168, 21271163, U1232211), the CAS/ SAFEA International Partnership Program for Creative Research Teams and CAS Hefei Science Center (2016HSC-IU011) and Fundamental Research Funds for the Central Universities (WK2060140021). The calculations were completed on the supercomputing system in the Supercomputing Center of USTC. The authors thank L. Shi for his help on the characterization of HRTEM. The authors thank beamline BL 14W1 (Shanghai Synchrotron Radiation Facility) for providing the beam time.

## Author contributions

Q.C. and J.S. designed and carried out research, analysed the data and wrote the paper. Y.Y. contributed to theoretical calculation in this paper and wrote the part of calculations. G.X., J.C. and P.J. co-wrote the manuscript. All authors discussed the results and commented on the manuscript.

## Additional information

**Competing interests:** The authors declare no competing financial interests.

**DOI: 10.1038/ncomms16028**  **OPEN**

# Corrigendum: Ruthenium-cobalt nanoalloys encapsulated in nitrogen-doped graphene as active electrocatalysts for producing hydrogen in alkaline media

Jianwei Su, Yang Yang, Guoliang Xia, Jitang Chen, Peng Jiang & Qianwang Chen

*Nature Communications* **8**:14969 doi: 10.1038/ncomms14969 (2017); Published 25 Apr 2017; Updated 20 Jun 2017

The legend to Fig. 8 of this Article contains a typographical error. The first sentence of this legend should read '(a) Co and (b) $Ru_3Co$ models.'

The Results section contains two typographical errors. The penultimate sentence of the first paragraph of the section 'Electrochemical characterization for HER catalysis' should read 'It was also found that S-4 with lower Ru content (3.58 wt.% Ru in RuCo@NC catalyst, obtained from ICP) showed better activity than the etched counterpart with higher Ru content (17.7 wt.% Ru in RuCo alloy, obtained from ICP), which was in good agreement with the above-mentioned trend of overpotentials.' In the same section, 'Especially, the S-4 catalyst exhibited an $ASA*C_S$ of $1.242\,\mathrm{mA\,cm^{-2}}$' should read 'Especially, the S-4 catalyst exhibited an $ASA*C_S$ of $1.242*C_S\,\mathrm{mA\,cm^{-2}}$'.

The legend to Supplementary Fig. 14 incorrectly states that the $\Delta j$ values plotted in h) correspond to those at 0.15 V *vs* RHE. The $\Delta j$ values in fact correspond to those at 0.1 V *vs* RHE.

DOI: 10.1038/ncomms16029    OPEN

# Erratum: Ruthenium-cobalt nanoalloys encapsulated in nitrogen-doped graphene as active electrocatalysts for producing hydrogen in alkaline media

Jianwei Su, Yang Yang, Guoliang Xia, Jitang Chen, Peng Jiang & Qianwang Chen

*Nature Communications* **8**:14969 doi: 10.1038/ncomms14969 (2017); Published 25 Apr 2017; Updated 20 Jun 2017

In the original version of this Article, which was received on 04 November 2016, the received date was incorrectly given as 04 November 2017. This has now been corrected in both the PDF and HTML versions of the Article.

