## [Peer Review File · Nature Communications]

Reviewers' comments:

Reviewer #1 (Remarks to the Author):

The authors reported RuCo nanoalloy@N doped graphene electrocatalysts for HER with excellent performance: low overpotentials of 28 mV at 10 mA/cm², low Tafel slope of 31 mV/dec and high stability, which is certainly interesting. A possible explanation was also proposed. Yet, some issues should be addressed before its acceptance. This manuscript is suitable for publication after minor corrections for the benefit of the reader.

1. Please add some characteristics to show the active sites and the active area;
2. The effect of the graphene's thickness on the catalysis has to be discussed? How to control the thickness?
3. The EIS for Co@C and CoRu@C should be added and fully discussed;
4. Why choose CoRu core, and not other cores?
5. Fig. 6 is not clear enough.

Reviewer #2 (Remarks to the Author):

This manuscript reports the utilization of Co-Ru alloys, with relatively low-loading of Ru, for the hydrogen evolution reaction (HER) in alkaline electrolyte. The authors performed synthesis, characterization, and electrochemical evaluation of 6 Co-Ru alloy catalysts; for benchmarking purpose the authors also compared their samples with the commercial Ru/C and Pt/C catalysts. The authors demonstrated that one of their catalysts showed higher activity and stability than the commercial catalysts. Furthermore, the authors performed DFT calculations in an attempt to understand the origin of the high HER activity of the Co-Ru catalysts. Overall the experimental results are convincing. The manuscript should be accepted for publication after the authors consider the following changes:

1. The DFT results are less convincing and self-inconsistent. From Table S1 (it is impossible to read Figure 6), the values for the ΔG of H is +0.31 eV for Ru₃Co, -0.33 eV for Ru(0001), and -0.39 eV for Pt(111). Based on the typical errors in DFT calculations, it is difficult to argue that the value of Ru₃Co is closer to zero than Pt. Using the same argument, the DFT results would suggest that Ru₃Co and Ru(001) should have similar HER activity, which is contradictory to their experimental data.
2. On the same topic, it is not very meaningful to compare RuCo nanoparticles with the Ru(001) and Pt(111) flat surfaces. The authors should perform DFT calculations on Ru and Pt clusters, with similar particle size as the RuCo cluster.
3. The title should be modified. The natural abundance of Ru is significantly less than Pt. The only reason that Ru is currently cheaper than Pt is because Ru is not used in large scale. If Ru is used as the catalysts for large scale HER applications, the cost of Ru would be much higher than Pt based on the fact that Ru is more scarce than Pt.
4. The authors need to carefully proof-read the manuscript. In several places "that" should be "than". The authors should also remove "we", "our", "can't", etc. in the main text.

Reviewer #3 (Remarks to the Author):

The authors report the synthesis of RuCo nano-alloy encapsulated in nitrogen doped graphene layers, which showed higher electrocatalytic hydrogen evolution reaction (HER) activity and

stability than that of the-state-of-the-art Pt catalyst in alkaline solutions. They conducted both experiments and computations to investigate the high activity. While the study fits well into the ongoing research trends to identify the alternatives of Pt for HER, there are some concerns regarding the level of study of the work.

1. For the activity comparison for different electrocatalysts, merely comparing the apparent overpotential is not sufficient, due to that different catalysts have various particle sizes, concentration of Ru, and density of the active sites. It is highly recommended that the authors conduct a series of activity normalization to take into account of the surface area, active site concentration, etc, to give a trend of different samples. Additionally, exchange current and turnover frequency need to be calculated to compare with those on Pt/C and other literature.

2. The title might be misleading, since the main component of the catalyst is Co but not Ru, even though adding Ru can largely increase the overall performance.

3. The biggest concern with regards to the computational part lies in the model construction of RuCo in N-graphene layers. According to Figure 2, the number of (nitrogen doped) graphene layers is 6-15, and within the computational models only one graphene layer was included. The authors cited ref 37 and 53 to support the single layer graphene model, however the sample synthesized in both literature are either single layer (ref 53) or 1-3 layers (ref 37).

Additionally, from the perspective of ΔGH^* , RuCo in nitrogen doped graphene (NG) shows a similar value with that on NG, the difference is only about 0.2 eV and could be taken within the error range. Take Pt for example, the value computed within this work is -0.39 eV and the well know value is -0.09 eV [Nørskov, J. K. et al. J. Electrochem. Soc. 152, J23–J26 (2005). Cited over 600 times].

There are many other factors to be considered about HER in alkaline solution, e.g. water splitting step, thermodynamics for Heyrovsky or Tafel step.

For Reviewer #1:

Remarks to the Author:

The authors reported RuCo nanoalloy@N doped graphene electrocatalysts for HER with excellent performance: low overpotentials of 28 mV at 10 mA/cm², low Tafel slope of 31 mV/dec and high stability, which is certainly interesting. A possible explanation was also proposed. Yet, some issues should be addressed before its acceptance. This manuscript is suitable for publication after minor corrections for the benefit of the reader.

Comments 1. Please add some characteristics to show the active sites and the active area;

Reply 1: Thank you for your advice, some previous works has reported the electrocatalysts of structures composed of alloy core encapsulated in nitrogen doped carbon. (ref 1-5) All of those results illustrated that the carbon atoms near the doping nitrogen on the graphene shell are the active sites, which is in good agreement with our calculation results.

We product new catalysts *via* the same procedures. The electrochemically active surface area (ECSA) of samples was usually estimated using a simple cyclic voltammetry method. (ref 6-7) The ECSA of a catalyst sample is calculated from the double layer capacitance according to following formula:

$$ECSA=C_{dl} / C_s$$

However, solving for the exact surface area of our material is difficult due to the unknown capacitive behavior (C_s) of the RuCo alloy electrode especially with N-doped carbon shell. But we can safely estimate relative surface areas of our seven samples, since the double layer capacitance (C_{dl}) is expected to be linearly proportional to effective active surface area for samples with similar composition and this method was also employed in some previous studies. (ref 4 and ref 8) The double layer capacitance is estimated by plotting the ΔJ at 0.1 V vs RHE against the scan rate, where the slope is twice of C_{dl} . The results showed that C_{dl} of S-0, S-1, S-2, S-3, S-4, S-5 and S-6 were 3.11, 10.27, 13.73, 16.61, 23.55, 10.95 and 10.16 mF/cm² respectively. This result also suggests that the catalytic performance of samples (S-0: 300 mV, S-1: 91 mV, S-2: 83 mV, S-3: 67 mV, S-4: 28 mV, S-5: 91 mV, S-6: 85 mV) increased with the increasing density of catalytical active sites.

Line 280 Page 12 and Line 281-286 Page 13 in the revised main article. Figure S14 in the revised supporting information.

Ref

- (1) Deng J., Ren P. J., Deng D.H., Yu L., Yang F. & Bao X. H. Highly active and durable non-precious-metal catalysts encapsulated in carbon nanotubes for hydrogen evolution reaction. *Energy Environ. Sci.* **7**, 1919-1923 (2014).
- (2) Deng J., Ren P. J., Deng D.H. & Bao X. H. Enhanced Electron Penetration through an Ultrathin Graphene Layer for Highly Efficient Catalysis of the Hydrogen Evolution Reaction. *Angew. Chem. Int. Ed.* **54**, 2100–2104 (2015).
- (3) Cui X.J., Ren P.J., Deng D.H., Deng J. & Bao X.H. Single layer graphene encapsulating non-precious metals as high-performance electrocatalysts for water oxidation. *Energy Environ. Sci.* **9**, 123-129 (2016).
- (4) Yang Y., Lun Z.Y., Xia G.L., Zheng F.C., He M.N. & Chen Q.W. Non-precious alloy encapsulated in nitrogen-doped graphene layers derived from MOFs as an active and durable hydrogen evolution reaction catalyst. *Energy Environ. Sci.* **8**, 3563-3571 (2015).
- (5) Yang Y. *et al.* Tuning Electronic Structures of Nonprecious Ternary Alloys Encapsulated in Graphene Layers for Optimizing Overall Water Splitting Activity. *ACS Catal.* **7**, 469–479 (2017).
- (6) Jin Y.S. *et al.* Porous MoO₂ Nanosheets as Non-noble Bifunctional Electrocatalysts for Overall Water Splitting. *Adv. Mater.* **28**, 3785-3790 (2016).
- (7) Tang, C. *et al.* Energy-Saving Electrolytic Hydrogen Generation: Ni₂P Nanoarray as a High-Performance Non-Noble-Metal Electrocatalyst. *Angew. Chem. Int. Ed.* **55**, 1-6 (2016).
- (8) Lukowski M.A., Daniel A.S., Meng F., Forticaux A., Li L. & Jin S. Enhanced Hydrogen Evolution Catalysis from Chemically Exfoliated Metallic MoS₂ Nanosheets. *J.Am.Chem.Soc.* **135**, 10274–10277 (2013).

Figure 1. Electrochemically active surface area measurements. (a-h) CV curves measured from 10 to 100 mV s^{-1} and corresponding Δj vs. scan rates plots of S-0, S-1, S-2, S-3, S-4, S-5 and S-6.

Comments 2. The effect of the graphene's thickness on the catalysis has to be discussed? How to control the thickness?

Reply 2: It is shown that the increasing thickness of the carbon shells may reduce the catalytic activity towards HER.(ref 1) In this work, the highest active electrocatalyst was synthesized at 600°C *via* one-step annealing of MOFs. In our electrocatalysts, most of nitrogen doped graphene are 6-15 layers, however, there still exists a large proportion of graphene shell of 1-5 layers (Fig. 2), which are good active sites for HER. The graphene with thinner layers will contribute more to the electrocatalytic performance via increase the charge transferring rate from alloy core to the graphene layer. Besides, the N- doped content and the alloy core also play the key role to electrocatalytic performance.

Line 164-165 Page 8 in the revised main article. Figure S8 in the revised supporting information.

Figure 2. High resolution transmission electron microscopy (HRTEM) of the S-4.

Ref

(1)Deng J., Ren P. J., Deng D.H. & Bao X. H. Enhanced Electron Penetration through an Ultrathin Graphene Layer for Highly Efficient Catalysis of the Hydrogen Evolution Reaction. *Angew. Chem. Int. Ed.* **54**, 2100–2104 (2015).

Comments 3. The EIS for Co@C and RuCo@C should be added and fully discussed;

Reply 3: Thanks for the reviewer's meaningful advice. Electrochemical impedance spectroscopy (EIS) Nyquist plots for Co@C and RuCo@C were collected in frequency range of $1-10^5$ Hz. (ref 1) The results suggest that Co@NC has a largest semicircle radius, indicative of a slightly higher charge-transfer impedance (9.32Ω) than that of S-1 (8.53Ω), of S-2 (8.55Ω), of S-3 (8.58Ω), of S-4, of S-5 (8.53Ω) and of S-6 (8.47Ω). (ref 2-3) Therefore, alloying Co with small amount of Ru could lead to a higher charge-transfer rate and more facile catalytic kinetics toward HER.

Figure 3. Electrochemical impedance spectroscopy (EIS) Nyquist plots for S-0, S-1, S-2, S-3, S-4, S-5 and S-6 collected in frequency range of $1-10^5$ Hz.

Line 295-300 Page 13 in the revised main article. Figure S15 in the revised supporting information.

Ref

- (1) Tang C. *et al.* Energy-Saving Electrolytic Hydrogen Generation: Ni₂P Nanoarray as a High-Performance Non-Noble-Metal Electrocatalyst. *Angew. Chem. Int. Ed.* **55**, 1-6 (2016).
- (2) Ma R.G., Zhou Y., Chen Y.F., Li P.X., Liu Q. & Wang J.C. Ultrafine Molybdenum Carbide Nanoparticles Compositing with Carbon as a Highly Active Hydrogen-Evolution Electrocatalyst. *Angew. Chem. Int. Ed.* **54**, 14723–14727 (2015).
- (3) Shi Z.P. *et al.* Porous nanoMoC@graphite shell derived from a MOFs-directed strategy: an efficient electrocatalyst for the hydrogen evolution reaction. *J. Mater. Chem. A* **4**, 6006–6013 (2016).

Comments 4. Why choose RuCo core, and not other cores?

Reply 4: This question is meaningful. We intend to develop cheap and robust electrocatalyst towards HER in basic media. Among many nonprecious materials, transition metals such as Fe, Co, and Ni and their alloys are regarded as potential substitutes for precious catalysts. (ref 1-4) In our previous work (ref 4), both the experimental results and the calculation results indicated that an increase in Co content in CoNiFe ternary alloys would improve the performance of HER in an alkaline solution. Therefore, we chose the Co-based alloy core. However, the TM-based catalysts still inferior to the Pt-based catalysts in overpotential and durability. Alloying noble metals with other transition metals is a possible route to prepare highly efficient catalysts with balance of good cost-competitiveness. In our previous works, we have produced electrocatalysts of PdCo and PtFeCo core for HER with high performance just in acid media. (ref 5-6) Metallic ruthenium is active for HER in basic media. (ref 7) Our group also produced Ru/MoO₂ electrocatalyst for HER with high performance, which has been submitted to JMCA. In fact, Ru is most economical in price among Pt-group metals, with one-twenty fifth price of Pt metal. Consequently, we choose RuCo alloy core in our electrocatalyst. Besides, alloying Co with small amount of Ru could lead to a lower charge-transfer impedance.

Besides, in previous work of JENS K. NØRSKOV (ref 8), they show schematically the calculated free energies of hydrogen adsorption on 736 distinct binary transition-metal surface alloys that can be formed from the 16 metals Fe, Co, Ni, Cu, As, Ru, Rh, Pd, Ag, Cd, Sb, Re, Ir, Pt, Au and Bi. Their calculation results clearly demonstrated that binary surface alloy of RuCo have high predicted activity for the HER ($|\Delta G_H|$ of RuCo: 0.1- 0.2 eV). This is another reason why we choose RuCo core.

Ref

- (1) Tavakkoli M. *et al.* Single-Shell Carbon-Encapsulated Iron Nanoparticles: Synthesis and High Electrocatalytic Activity for Hydrogen Evolution Reaction. *Angew. Chem. Int. Ed.* **54**, 4535-4538 (2015).

- (2) Deng J., Ren P. J., Deng D.H. & Bao X. H. Enhanced Electron Penetration through an Ultrathin Graphene Layer for Highly Efficient Catalysis of the Hydrogen Evolution Reaction. *Angew. Chem. Int. Ed.* **54**, 2100–2104 (2015).
- (3) Wang, J. *et al.* Cobalt nanoparticles encapsulated in nitrogen-doped carbon as a bifunctional catalyst for water electrolysis. *J. Mater. Chem. A* **2**, 20067-20074 (2014).
- (4) Yang Y. *et al.* Tuning Electronic Structures of Nonprecious Ternary Alloys Encapsulated in Graphene Layers for Optimizing Overall Water Splitting Activity. *ACS Catal.* **7**, 469–479 (2017).
- (5) Chen J.T., Yang Y., Su J.W., Jiang P., Xia G.L. & Chen Q.W. Enhanced activity for Hydrogen Evolution Reaction over CoFe Catalysts by Alloying with small amount of Pt. *ACS Appl. Mater. Inter.* **DOI: 10.1021/acsami.6b12065** (2017).
- (6) Chen J. T. *et al.* Active and Durable Hydrogen Evolution Reaction Catalyst Derived from Pd-Doped Metal-Organic Frameworks. *ACS Appl. Mater. Interfaces* **8**, 13378-13383 (2016).
- (7) Zheng Y. *et al.* High Electrocatalytic Hydrogen Evolution Activity of an Anomalous Ruthenium Catalyst. *J. Am. Chem. Soc.* **138**, 16174-16181 (2016).
- (8) Greeley J., Jaramillo T. F., Bonde J., Chorkendorff I. & Nørskov J. K. Computational high-throughput screening of electrocatalytic materials for hydrogen evolution. *Nature Mater.* **5**, 909-913 (2006).

Comments 5. Fig. 6 is not clear enough.

Reply 5: The above picture have been replaced by a new clear picture in the manuscript.

Picture 6 in the revised main article.

For Reviewer #2:

Remarks to the Author:

This manuscript reports the utilization of Co-Ru alloys, with relatively low-loading of Ru, for the hydrogen evolution reaction (HER) in alkaline electrolyte. The authors performed synthesis, characterization, and electrochemical evaluation of 6 Co-Ru alloy catalysts; for benchmarking purpose the authors also compared their samples with the commercial Ru/C and Pt/C catalysts. The authors demonstrated that one of their catalysts showed higher activity and stability than the commercial catalysts. Furthermore, the authors performed DFT calculations in an attempt to understand the origin of the high HER activity of the Co-Ru catalysts. Overall the experimental results are convincing. The manuscript should be accepted for publication after the authors consider the following changes:

Comments 1. The DFT results are less convincing and self-inconsistent. From Table S1 (it is impossible to read Figure 6), the values for the ΔG of H is +0.31 eV for Ru₃Co, -0.33 eV for Ru(0001), and -0.39 eV for Pt(111). Based on the typical errors in DFT calculations, it is difficult to argue that the value of Ru₃Co is closer to zero than Pt. Using the same argument, the DFT results would suggest that Ru₃Co and Ru(001) should have similar HER activity, which is contradictory to their experimental data.

Comments 2. On the same topic, it is not very meaningful to compare RuCo nanoparticles with the Ru(001) and Pt(111) flat surfaces. The authors should perform DFT calculations on Ru and Pt clusters, with similar particle size as the RuCo cluster.

Reply to question 1 and 2: Thank you for your suggestions. We have rebuilt and optimized Pt and Ru cluster with the same atom number (55) and calculated parameters as RuCo alloys which was also suggested by reviewer 2. Besides, some possible H* adsorption sites such as top, bridge, hcp and hollow sites are taken into considerations in our cluster models (As

shown in Figure 5 and Figure 6). It should be noted that, during optimization process the hollow site of Pt55 and top site of Ru55 are tended to be bridge and hollow sites, respectively, indicating the latter are more moderate adsorption sites. The calculated ΔG_H^* of different models and sites are illustrated in Table 1.

The calculated results show Ru55 cluster exhibited even better ΔG_H^* than RuCo and Pt55, which is not agreement with our previous experimental results based on purchased Ru powder. Unlike metal surface, metal cluster exhibits more high active sites such as facets, corners, edges and usually corresponds to experimental nano-sized metal particles. However, the Ru powder measured before was purchased from manufacturer and the size of them was in micron order as illustrated in Figure 4a. Therefore, we synthesized nano-sized Ru nanoparticles by ourselves through RuCl₃ and NaBH₄ as Precursors (detailed information is in the supplement information). The performance of nano-sized Ru catalyst was even a litter better than our S-4 at high overpotential, which was in good agreement with our calculated cluster result. For Pt cluster, even its calculated result was as good as our RuCo models, however, the measured activity was inferior to our S-4 sample. Similar results also existed in a recently published research (ref 1), Qiao and his co-workers have proved that, apart from the ΔG_H^* , the water dissociation kinetics would also affect the overall reaction rate, especially under basic electrolyte. According to their calculated results, when the kinetics of water dissociation from the Volmer step is considered, Pt exhibits a significant higher energy barrier than Ru and others, indicating sluggish water dissociation during catalytic process. Therefore, from the kinetic viewpoint, conversely, Pt didn't demonstrate as good catalytic performance as depicted by calculated ΔG_H^* , which might explain the inferior measured activity than our RuCo alloys. We also tried to calculate energy barrier of water dissociation process for our models, however, due to the very large models (C₂₄₀Co₅₂Ru₃ containing 295 atoms), it's very difficult and time-consuming to search for the exact transition state and energy barrier of ours. However, combing our calculated ΔG_H^* results with Qiao's research, it's sufficient to give a reasonable explanation. Therefore, both the experimental and the calculated results indicated that alloying Co with small amount of Ru could obtain highly active eletrocatalyst comparable to pure Ru catalyst.

Figure 4 (a) The FESEM image of the commercial Ru powder catalysts. (b and c) The FESEM and TEM images of our obtained nano-sized Ru. (d-f) HER polarization curves and the Tafel plot of Nano Ru.

Figure 5. (a) Pt_{55} cluster model from side view, (b-d) H^* adsorbed on top, bridge, and hollow sites on cluster from top view, respectively.

Figure 6. (a) Ru₅₅ cluster model from side view, (b-e) H* adsorbed on top, bridge, hollow and hcp sites on cluster from top view, respectively.

Table 1. Calculated ΔGH^* of different adsorption sites on Pt₅₅ and Ru₅₅.

Models/Sites	Top	Bridge	Hollow	HCP
Pt ₅₅	-0.30	-0.37	-0.37	-
Ru ₅₅	-0.09	0.03	-0.09	-0.07

Line 354 Page 16, Line 355-376 Page 17, Line 377-384 Page 18 and Figure 6 in the revised main article. Page 2, Figure S18, Figure S19, Figure S20 and Table S4 in the revised supporting information.

Ref

(1) Zheng Y. *et al.* High Electrocatalytic Hydrogen Evolution Activity of an Anomalous Ruthenium Catalyst. *J. Am. Chem. Soc.* **138**, 16174-16181 (2016).

Comments 3. The title should be modified. The natural abundance of Ru is significantly less than Pt. The only reason that Ru is currently cheaper than Pt is because Ru is not used in large scale. If Ru is used as the catalysts for large scale HER applications, the cost of Ru would be much higher than Pt based on the fact that Ru is more scarce than Pt.

Reply 3: Thank you for pointing out this problem. The title has been revised. The revised title is that "RuCo nanoalloys Encapsulated in N-doped graphene layers as a cheaper alternative to Pt-based electrocatalysts for HER in alkaline media".

Line 1-2 page 1 in the revised main article.

Comments 4. The authors need to carefully proof-read the manuscript. In several places "that" should be "than". The authors should also remove "we", "our", "can't", etc. in the main text.

Reply4: The above mistakes have been revised in the manuscript. And the expression of English has been carefully checked and revised for the manuscript.

For Reviewer #3:

The authors report the synthesis of RuCo nano-alloy encapsulated in nitrogen doped graphene layers, which showed higher electrocatalytic hydrogen evolution reaction (HER) activity and stability than that of the-state-of-the-art Pt catalyst in alkaline solutions. They conducted both experiments and computations to investigate the high activity. While the study fits well into the ongoing research trends to identify the alternatives of Pt for HER, there are some concerns regarding the level of study of the work.

Comments 1. For the activity comparison for different electrocatalysts, merely comparing the apparent overpotential is not sufficient, due to that different catalysts have various particle sizes, concentration of Ru, and density of the active sites. It is highly recommended that the authors conduct a series of activity normalization to take into account of the surface area, active site concentration, etc, to give a trend of different samples. Additionally, exchange current and turnover frequency need to be calculated to compare with those on Pt/C and other literature.

Reply 1: Thank you for your advice, we have carried out the statistical analysis of the particle sizes of alloy core. Obviously, all the particles have the similar mean diameter of near 30 nm.

The electrochemically active surface area (ECSA) of samples was usually estimated using a simple cyclic voltammetry method. (ref 1-2) The ECSA of a catalyst sample is calculated from the double layer capacitance according to following formula: $ECSA = C_{dl} / C_s$. However, solving for the exact surface area of our material is difficult due to the unknown capacitive behavior (C_s) of the RuCo alloy electrode especially with nitrogen doped carbon shell. But we can safely estimate relative surface areas of seven samples, since the double layer capacitance (C_{dl}) is expected to be linearly proportional to effective active surface area for samples with similar composition and this method was also employed in previous study. (ref 3-4) The double layer capacitance is estimated by plotting the ΔJ at 0.1 V vs RHE against the scan rate, where the slope is twice of C_{dl} . The results showed that C_{dl} of S-0, S-1, S-2, S-3, S-4, S-5 and S-6 were 3.11, 10.27, 13.73, 16.61, 23.55, 10.95 and 10.16 mF/cm² respectively. This result also suggests that the catalytic performance of samples (S-0: 300 mV, S-1: 91 mV, S-2: 83 mV, S-3: 67 mV, S-4: 28 mV, S-5: 91 mV, S-6: 85 mV) increased with the increasing density of catalytically active sites.

The BET surface area and pore size distribution of the seven samples show similar adsorption/desorption isotherm curves and pore size distribution. The BET surface area were listed in Table 2. Obviously, the BET surface area of samples are not linear with the catalytic performance. Therefore, the effect of surface area on activity is negligible in our materials compared with other influencing factors.

A series of activity normalization of special activity (SA) and active site activity (ASA), taking into account of the surface area and active site concentration respectively at overpotential of 100 mV, were shown in Fig. 15. (ref 5) Especially, the S-4 catalyst exhibited a SA of 0.707 mA cm⁻², which are much higher than other catalysts. Besides, the result also suggests that the SA of samples increased with the increasing catalytic active in general, suggesting the effect of surface area is a negligible factor on activity. Due to the unknown capacitive behavior (C_s) of the RuCo alloy electrode especially with nitrogen doped carbon shell, we carried out activity normalization of $ASA * C_s$. Especially, the S-4 catalyst exhibited an $ASA * C_s$ of 1.242 mA cm⁻², which are much higher than other catalysts, which have the similar values of $ASA * C_s$.

The exchange current density (j_0) values obtained from Tafel plots. (ref 6-7) As read from Figure 16, the trend of j_0 is S-4 ($10^{-2.48}$) > Ru powder ($10^{-2.74}$) > Pt/C ($10^{-2.94}$). Due to the capacitive behavior (C_s) of the RuCo alloy electrode is unknown, we can't obtain the values of electrochemically active surface area. Therefore, the calculation of turnover frequency is beyond our abilities right now.

Figure 7. (e, f) Statistical analysis of the particle sizes of Co metal in S0.

Figure 8. (e, f) Statistical analysis of the particle sizes of RuCo metal in S1.

Figure 9. (e, f) Statistical analysis of the particle sizes of RuCo metal in S2.

Figure 10. (e, f) Statistical analysis of the particle sizes of RuCo metal in S3.

Figure 11. (e, f) Statistical analysis of the particle sizes of RuCo metal in S5.

Figure 12. (e, f) Statistical analysis of the particle sizes of RuCo metal in S6.

Figure 13. (a,b) Statistical analysis of the particle sizes of RuCo metal in S4.

Figure 14. N_2 adsorption–desorption isotherm and pore size distribution plot (inset) of S-0, S-1, S-2, S-3, S-5 and S-6.

Figure 1. Electrochemically active surface area measurements. (a-h) CV curves measured from 10 to 100 mV s⁻¹ and corresponding Δj vs. scan rates plots of S-0, S-1, S-2, S-3, S-4, S-5 and S-6.

Figure 15. Activity normalization of (a) special activity (SA) and (b) active site activity (ASA) taking into account of the surface area and active site concentration respectively at overpotential of 100 mV.

Figure 16. the exchange current density of S-4, Ru and Pt/C.

Table 2. Mean particle size, specific surface area and double layer capacitance of various catalysts.

Catalyst	S-0	S-1	S-2	S-3	S-4	S-5	S-6
Particle size(nm)	30.87	28.09	28.75	28.21	28.38	29.94	28.28
Specific surface area(m ² g ⁻¹)	46.49	47.41	42.87	37.68	41.37	50.77	44.27
Double layer	3.11	10.27	13.73	16.61	23.55	10.95	10.16

capacitance(mF/cm ²)							
--	--	--	--	--	--	--

Line 231-232 Page 10, Line 233 Page 11, Line 280 Page 12, Page 281-295 Page 13 and Page 311-314 Page 14 in the revised main article. Figure S1-6 e and f, Figure S7, Figure S8, Figure S11, Figure S14, Figure S17 and Table S1 in the revised supporting information.

Ref

- (1) Jin Y.S. *et al.* Porous MoO₂ Nanosheets as Non-noble Bifunctional Electrocatalysts for Overall Water Splitting. *Adv. Mater.* **28**, 3785-3790 (2016).
- (2) Tang, C. *et al.* Energy-Saving Electrolytic Hydrogen Generation: Ni₂P Nanoarray as a High-Performance Non-Noble-Metal Electrocatalyst. *Angew. Chem. Int. Ed.* **55**, 1-6 (2016).
- (3) Lukowski M.A., Daniel A.S., Meng F., Forticaux A., Li L. & Jin S. Enhanced Hydrogen Evolution Catalysis from Chemically Exfoliated Metallic MoS₂ Nanosheets. *J.Am.Chem.Soc.* **135**, 10274–10277 (2013).
- (4) Yang Y., Lun Z.Y., Xia G.L., Zheng F.C., He M.N. & Chen Q.W. Non-precious alloy encapsulated in nitrogen-doped graphene layers derived from MOFs as an active and durable hydrogen evolution reaction catalyst. *Energy Environ. Sci.* **8**, 3563-3571 (2015).
- (5) Zhu Y.L., Zhou W. Yu J., Chen Y.B., Liu M.L. & Shao Z.P. Enhancing Electrocatalytic Activity of Perovskite Oxides by Tuning Cation Deficiency for Oxygen Reduction and Evolution Reactions. *Chem. Mater.* **28**, 1691-1697 (2016).
- (6) Jiao Y., Zheng Y., Jaroniec M. & Qiao S.Z. Design of electrocatalysts for oxygen- and hydrogen-involving energy conversion reactions. *Chem. Soc. Rev.* **44**, 2060—2086 (2015).
- (7) Zheng Y. *et al.* High Electrocatalytic Hydrogen Evolution Activity of an Anomalous Ruthenium Catalyst. *J. Am. Chem. Soc.* **138**, 16174-16181 (2016).

Comments 2. The title might be misleading, since the main component of the catalyst is Co but not Ru, even though adding Ru can largely increase the overall performance.

Reply 2: Thank you for pointing out this problem. The title has been revised. The revised title is that “RuCo nanoalloys Encapsulated in N-doped graphene layers as a cheaper alternative to Pt-based electrocatalysts for HER in alkaline media”.

Line 1-2 page 1 in the revised main article.

Comments 3. The biggest concern with regards to the computational part lies in the model construction of RuCo in N-graphene layers. According to Figure 2, the number of (nitrogen doped) graphene layers is 6-15, and within the computational models only one graphene layer was included. The authors cited ref 37 and 53 to support the single layer graphene

model, however the sample synthesized in both literature are either single layer (ref 53) or 1-3 layers (ref 37).

Reply 5: Yes, we appreciate your comment. Indeed, most of nitrogen doped graphene are 6-15 layers in our catalysts. There still exist a large proportion of graphene shell (less than five layers) in the composites, which could be seen in the images of Fig. 2. Besides, a broad and weak second-order band observed at approximately $2,700\text{ cm}^{-1}$ in fig. 3b and the shape of the D and 2D bands are characteristic features of graphene with several layers. (ref 1)

Ref 37 in main article clearly illustrated that the electron of an alloy cluster can penetrate through three graphene layers. In fact, the model for calculation is usually simplified compared to real structure in order to reduce calculation, meanwhile metallic core encapsulated with carbon shell composite structure have been reported in previous works using simplified graphene encapsulated calculation models.(ref.3-5) Previous works (ref 3-5) have proved that the calculated results using only one graphene layer model were in good agreement with the experimental performance of the alloys with about three to five layer graphene shell. In our sample, there is still a large proportion of graphene shell of 1-5 layers, which might be good active sites for HER. Therefore, the simplified one layer graphene model here could provide valuable information to get insight into the catalytic mechanism.

Figure 2. High resolution transmission electron microscopy (HRTEM) of the S-4.

Line 164-165 Page 8 in the revised main article. Figure S8 in the revised supporting information.

Ref

- (1) Ferrari A. C. & Basko D.M., Raman spectroscopy as a versatile tool for studying the properties of graphene. *Nat. nanotechnol.* **8**, 235-246 (2013).
- (2) Deng J., Ren P. J., Deng D.H. & Bao X. H. Enhanced Electron Penetration through an Ultrathin Graphene Layer for Highly Efficient Catalysis of the Hydrogen Evolution Reaction. *Angew. Chem. Int. Ed.* **54**, 2100–2104 (2015).
- (3) Deng J., Ren P. J., Deng D.H., Yu L., Yang F. & Bao X. H. Highly active and durable non-precious-metal catalysts encapsulated in carbon nanotubes for hydrogen evolution reaction. *Energy Environ. Sci.* **7**, 1919-1923 (2014).
- (4) Yang Y., Lun Z.Y., Xia G.L., Zheng F.C., He M.N. & Chen Q.W. Non-precious alloy encapsulated in nitrogen-doped graphene layers derived from MOFs as an active and durable hydrogen evolution reaction catalyst. *Energy Environ. Sci.* **8**, 3563-3571 (2015).

(5) Yang Y. *et al.* Tuning Electronic Structures of Nonprecious Ternary Alloys Encapsulated in Graphene Layers for Optimizing Overall Water Splitting Activity. *ACS Catal.* **7**, 469–479 (2017).

Comments 4. Additionally, from the perspective of ΔG_{H^*} , RuCo in nitrogen doped graphene (NG) shows a similar value with that on NG, the difference is only about 0.2 eV and could be taken within the error range. Take Pt for example, the value computed within this work is -0.39 eV and the well know value is -0.09 eV [Nørskov, J. K. *et al.* *J. Electrochem. Soc.* **152**, J23–J26 (2005). Cited over 600 times].

There are many other factors to be considered about HER in alkaline solution, e.g. water splitting step, thermodynamics for Heyrovsky or Tafel step.

Reply 4: Thanks for your suggestion. The calculated method and value of ΔG_{H^*} for Pt (111) surface provided by Nørskov, J. K. is a standard and commonly used by other researches. However, the exact value of ΔG_{H^*} is usually varied from each other due to the difference of geometry models, calculating parameters, adsorption sites and so on among different articles (ref 1,2). Therefore, it's more reasonable to compare the result based on same or uniform models, calculating parameters and so on. We have rebuilt and optimized Pt and Ru cluster with the same atom number (55) and calculated parameters as RuCo alloys which was also suggested by reviewer 2. Besides, some possible H^* adsorption sites such as top, bridge, hcp and hollow sites are taken into considerations in our cluster models (As shown in Figure 5 and Figure 6). It should be noted that, during optimization process the hollow site of Pt_{55} and top site of Ru_{55} are tended to be bridge and hollow sites, respectively, indicating the latter are more moderate adsorption sites. The calculated ΔG_{H^*} of different models and sites are illustrated in Table 1.

The calculated results show Ru_{55} cluster exhibited even better ΔG_{H^*} than RuCo and Pt_{55} , which is not agreement with our previous experimental results based on purchased Ru powder. Unlike metal surface, metal cluster exhibits more high active sites such as facets, corners, edges and usually corresponds to experimental nano-sized metal particles. However, the Ru powder measured before was purchased from manufacturer and the size of them was in micron order as illustrated in Figure 4a. Therefore, we synthesized nano-sized Ru nanoparticles by ourselves through $RuCl_3$ and $NaBH_4$ as Precursors (detailed information is in the supplement information). The performance of nano-sized Ru catalyst was even a litter better than our S-4 at high overpotential, which was in good agreement with our calculated cluster result. For Pt cluster, even its calculated result was as good as our RuCo models, however, the measured activity was inferior to our S-4 sample. Similar results also existed in a recently published research (ref 1), Qiao and his co-workers have proved that, apart from the ΔG_{H^*} , the water dissociation kinetics would also affect the overall reaction rate, especially under basic electrolyte. According to their calculated results, when the kinetics of water dissociation from the Volmer step is considered, Pt exhibits a significant higher energy barrier than Ru and others, indicating sluggish water dissociation

during catalytic process. Therefore, from the kinetic viewpoint, conversely, Pt didn't demonstrate as good catalytic performance as depicted by calculated ΔG_{H^*} , which might explain the inferior measured activity than our RuCo alloys. We also tried to calculate energy barrier of water dissociation process for our models, however, due to the very large models ($C_{240}Co_{52}Ru_3$ containing 295 atoms), it's very difficult and time-consuming to search for the exact transition state and energy barrier of ours. However, combing our calculated ΔG_{H^*} results with Qiao's research, it's sufficient to give a reasonable explanation. Therefore, both the experimental and the calculated results indicated that alloying Co with small amount of Ru could obtain highly active electrocatalyst comparable to pure Ru catalyst.

Figure 4 (a) The FESEM image of the commercial Ru powder catalysts. (b and c) The FESEM and TEM images of our obtained nano-sized Ru. (d-f) HER polarization curves and the Tafel plot of Nano Ru.

Figure 5. (a) Pt₅₅ cluster model from side view, (b-d) H* adsorbed on top, bridge, and hollow sites on cluster from top view, respectively.

Figure 6. (a) Ru₅₅ cluster model from side view, (b-e) H* adsorbed on top, bridge, hollow and hcp sites on cluster from top view, respectively.

Table 1. Calculated ΔGH^* of different adsorption sites on Pt₅₅ and Ru₅₅.

Models/Sites	Top	Bridge	Hollow	HCP
Pt ₅₅	-0.30	-0.37	-0.37	-
Ru ₅₅	-0.09	0.03	-0.09	-0.07

Line 354 Page 16, Line 355-376 Page 17, Line 377-384 Page 18 and Figure 6 in the revised main article. Page 2, Figure S18, Figure S19, Figure S20 and Table S4 in the revised supporting information.

Ref

(1) Zheng Y. *et al.* High Electrocatalytic Hydrogen Evolution Activity of an Anomalous Ruthenium Catalyst. *J. Am. Chem. Soc.* **138**, 16174-16181 (2016).

REVIEWERS' COMMENTS:

Reviewer #1 (Remarks to the Author):

The authors have improved the article. It is ready for publication.

Reviewer #2 (Remarks to the Author):

The authors have addressed my questions.

Reviewer #3 (Remarks to the Author):

The authors have carefully addressed the comments, as well as carried out additional analysis and computation works. The quality of the article is greatly improved and is suggested to be accepted. Please carefully check the spelling in the manuscript, for example line 217 'Therefoourre' should be 'Therefore'.

Additionally, in Figure S14 (f), name of x-axis should keep consistent with other panels, to read Potential (V versus RHE), rather than Overpotential (V).

Reviewers' comments:

For Reviewer #1:

Remarks to the Author:

The authors have improved the article. It is ready for publication.

For Reviewer #2:

Remarks to the Author:

The authors have addressed my questions.

For Reviewer #3:

Remarks to the Author:

The authors have carefully addressed the comments, as well as carried out additional analysis and computation works. The quality of the article is greatly improved and is suggested to be accepted. Please carefully check the spelling in the manuscript, for example line 217 'Therefooure' should be 'Therefore'.

Additionally, in Figure S14 (f), name of x-axis should keep consistent with other panels, to read Potential (V versus RHE), rather than Overpotential (V).

Reply: The above mistakes have been revised in the manuscript. And the spelling of English has been carefully checked and revised for the manuscript.